# Study on the dynamic distribution of spores of powdery mildew pathogen in wheat by rotor airflow of plant protection UAV

**Weicai Qin**[1,2]*, **Panyang Chen**[3], **Ruyan He**[4]

**1** Suzhou Polytechnic Institute of Agriculture, Suzhou, China, **2** Nanjing Institute of Agricultural Mechanization, Ministry of Agriculture and Rural Affairs, Nanjing, China, **3** Nanjing Institute of Technology, Nanjing, China, **4** Shanghai Ocean University, Shanghai, China

* qinweicai@caas.cn

## Abstract

Plant protection drones are fast and efficient application machines that are characterised by high application efficiency and no damage to crops. They are particularly suitable for small areas of farmland and mountainous terrain in regions such as Asia and are currently the dominant insecticide application technology in China. The presence of wind is a prerequisite for the spread and dissemination of airborne diseases and it can directly influence the distance and height of ascent of pathogenic spores. This paper investigates the effect of downwash airflow generated by the flight altitude of a plant protection drone on the horizontal distribution, vertical distribution and ground distribution of powdery mildew spores in wheat. Monitoring the changing dynamics of airborne powdery mildew conidia using spore traps. The test results show that: the number of powdery mildew pathogenic spores is related to various factors such as weather, relative humidity and wind speed; the release of spores is greatly influenced by airflow disturbances but has little effect at the early stages of sporulation; the disease is caused by the accumulation process of pathogenic spores and in the control of powdery mildew in wheat, preventive spraying should be carried out within 2–3 days of the germination of pathogenic spores. The study lays the foundation for further in-depth research on the spread of powdery mildew spores and improved pest control, and provides a basis for scientific and rational spraying and control by agricultural drones.











**Data Availability Statement:** All relevant data are within the paper.

**Funding:** This research was funded by the National Natural Science Foundation of China (Grant No.31971804); Jiangsu Modern Agricultural

## 1. Introduction

Wheat powdery mildew caused by the biotrophic fungi Blumeria graminis (DC) Speer f. sp. tritici Emend. E.J. Marshcl, is the most important wheat diseases worldwide, which occurs in regions with maritime and semi-continental climates and causes severe yield losses. including in Gansu, Jiangsu Province, China [1, 2]. Especially in the heading stage of wheat, powdery mildew occurs seriously in the lower part of the canopy. At this time, the leaves of the wheat canopy cover each other, which makes it difficult for ground equipment to work, or causes great damage to the wheat due to rolling. In addition, it is difficult for the artificial spraying

Machinery Equipment and Technology Demonstration and Promotion Project (Grant No. NJ2022-17); Suzhou Agricultural Independent Innovation Pro-ject (SNG2022061); Suzhou Agricultural Vocational and Technical College Landmark Achievement Cultivation Project (CG [2022]02); Soft Science Project of Suzhou Association for Science and Technology. The funders had no role in study design, data collection and analysis, decision to publish, or preparation of the manuscript.

**Competing interests:** The authors have declared that no competing interests exist.

liquid to enter the middle and lower areas of wheat, which often leads to outbreaks of wheat powdery mildew, and conidia spread everywhere, which seriously affects the high and stable yield of wheat [3–5]. In the later stage of wheat growth, it is difficult to walk and labor-intensive when spraying pesticides. Large-volume spraying not only wastes pesticides, but also causes serious harm to pesticide application personnel and the ecological environment [6–8]. Moreover, there is a lack of emergency prevention and control capabilities for outbreaks of diseases and insect pests, and it is impossible to prevent and control diseases and insect pests in time, thus delaying the opportunity and causing serious losses. Therefore, it is urgent to solve the current situation of low level of mechanization of wheat pest control in China, enhance the ability to prevent and control sudden large-scale pests and diseases, and alleviate the shortage of rural labor [9–13].

Aerial application, commonly called crop dusting, involves spraying crops with fertilizers, pesticides, fungicides, and other crop protection materials from agricultural aircraft [14]. Farmers in Japan and South Korea have relatively small arable land per household, with mountainous terrain and small arable land, which are not suitable for manned fixed-wing aircraft operations. Therefore, small unmanned helicopters are the main agricultural aviation in this area. Compared with large agricultural aircraft, small unmanned helicopters have their special advantages, no need for special take-off and landing airports, good maneuverability, etc [15–18], and can have better terrain adaptability and low-altitude spray capabilities [19–23]. In recent years, China's aviation plant protection machinery has developed rapidly, especially small agricultural unmanned helicopter aerial spray [24–26]. In view of the current situation of the development of small unmanned helicopters in China, scholars have carried out an exploration of field spraying experiments [27]. For example, Qiu (2013) [28] studied the distribution law of droplet deposition under different flight heights and flight speeds of the CD-10 unmanned helicopter, and established a relationship model between deposition concentration, deposition uniformity, flight speed, flight height and the interaction between the two factors. In agricultural pest control, Xue, et al. (2013) [29] studied the control effect of N-3 unmanned helicopter on rice planthopper and rice leaf roller, pointing out that compared with the traditional field manual application, the unmanned helicopter operation efficiency can be increased by 60 times, the active ingredient of the spraying liquid can be reduced by 20%-30%, and the labour intensity is greatly reduced, providing a working platform for rapid and effective prevention and control of outbreaks of diseases, pests and weeds in paddy fields and promoting the upgrading of large-scale plant protection technology of rice. Qin, et al. (2016) [30] studied the effect of HyB-15L UAV at different working heights and speeds on the deposition and distribution of droplets and the control effect of rice planthoppers.

The presence of wind is a prerequisite for the spread and dissemination of airborne diseases, which can directly affect the diffusion distance and rising height of pathogen spores. The release of spores requires not only overcoming its attachment to the spore-forming organ, but also passing through and out of the still air layer of the host's surface. However, the release rate of spores is not linear with the wind speed. Generally speaking, the spores can be released only when the wind speed reaches a certain level. The analysis shows that once the wind speed reaches this speed, almost all the spores that can be released are released around this moment. For example, the direct wind speed of the maize small spot fungus conidia themselves is 5 $m\cdot s^{-1}$, but because of the interfacial layer effect on the leaf surface, and the closer the leaf surface, the smaller the wind speed, so the wind speed is projected to reach 25 $m\cdot s^{-1}$ to make the leaf surface wind speed reach 5 $m\cdot s^{-1}$ (Force equivalent to 2000 times the weight of the spore). However, in practice it was observed that the average wind speed did not reach 25 $m\cdot s^{-1}$ and that the maize microspot conidia had been released due to gusts of wind (Aylor, 1978) [31, 32]. Thus gusts play a greater role than average wind speed in spore release, and the greater the

number of gusts, the greater the amount of spores released, showing that gusts play a special role in spore release. Thus gusts play a greater role than average wind speed in spore release, and the greater the number of gusts, the greater the amount of spores released, showing that gusts play a special role in spore release. However, some pathogens require little force to release, such as powdery mildew, and the release of large numbers of spores can be accelerated when the wind shakes the leaves. In addition, wind has a strong influence on the spread of pathogens, with the amount of wind speed directly related to the distance the pathogens can travel [33]. However, there is a lack of in-depth research on the effect of under-rotor wash airflow on pathogens from plant protection drones.

Powdery mildew of wheat is an airborne disease caused by the obligate parasitic fungus B. graminis f.sp. tritici [34], The speed of the rotor airflow generated during the operation of the plant protection UAV reaching the crop canopy is generally greater than 3 $m \cdot s^{-1}$ while the rice planthopper and powdery mildew spores are easily affected by the airflow and spread around, In the study, it was found that after the plant protection drone flew over the rice crops, there would be obvious rice planthopper flying phenomenon. Similarly, when using plant protection drones to control rape sclerotinia, it was found that the disease became more serious after spraying. Therefore, it may be that the air flow of the plant protection drones caused the sclerotia in the mature stage to spread around, aggravating the incidence of sclerotinia. In view of the above situation, this chapter analyzes the influence of the downwash air flow generated by the rotor on the horizontal, vertical and ground distribution of rice canopy rice planthopper and wheat powdery mildew spores by studying the flying height of the plant protection UAV. Using sticky insect boards and spore traps to monitor the changing dynamics of rice planthoppers and powdery mildew conidia in crop canopy and air, and analyze the impact of environmental factors on the production of powdery mildew conidia [35–37]. To clarify the influence of drone operation height on the distribution of rice planthopper and powdery mildew conidia in the crop canopy, it will lay a foundation for further in-depth research on the occurrence of this disease and pest control in the future, In order to reduce the amount and frequency of pesticide use, it provides a basis for scientific and rational application of agricultural drones.

## 2. Materials and methods

### 2.1. Test site conditions and test time

The 2020 trial was conducted in the experimental field of Hongze Lake Farm, Suqian, Jiangsu. Tobacco farmer No.19 was selected for the study, with a plant spacing of 10 cm × 20 cm and a plant height of 0.8–0.95 m. The trial was conducted from 20 to 27 April. The growing period is tassel-filling with consistent growth and the disease is wheat powdery mildew.

### 2.2. Test apparatus and equipment

XSP-6C Biological Microscope; GM8901 Mini Meteorological Measuring Instrument (Shenzhen Wanyitong Instrument Trading Company); The UAV platforms are the N-3 and the HYB-15L, Specific parameters are shown in Table 1.

### 2.3. Experimental design

**2.3.1. Cell arrangements.** A plot (40 m × 100 m) with the same crop variety and growth, and a regular field size was selected as the test site and divided into three areas, recorded as A, B and C. Area A was the N-3 plant protection drone test area, with the flight height set at 5 m and flight speed at 4 $m \cdot s^{-1}$ respectively; Area B was the HYB-15L plant protection drone test area, with the flight height set at 1.5 m and flight speed at 4 $m \cdot s^{-1}$ respectively; Area C was the

Table 1. N-3 and HYB-15L unmanned aircraft parameters.

| Model / Parameters | N-3 | HYB-5 |
|---|---|---|
| Fly speed/$(m{\cdot}s^{-1})$ | 3 | 3–5 |
| Fly height/m | 5–7 | 0.5–1.5 |
| Main rotor diameter/mm | 3115 | 2143 |
| Size /mm | 2696/720/1109 | 1955/455/625 |
| Takeoff weight/kg | 100 | 30.80 |
| Rotating speed/rpm | 830 | 1400 |
| Number of blades/pcs | 2 | 1 |


control area. There was a 30 m wide separation area between the test areas. Two TPBZ3 spore traps were placed in the middle of each test area, with the inlets of the two traps located at 0.5 m and 1.5 m from the ground respectively, and the two traps were 10 m apart, as shown in Fig 1.

**2.3.2. Aerial spore capture and observation.** Catching began on 20 April, with the machine being switched on at 1:00 pm for 6 hours each day as the wheat flour fungus is at its peak spore production from 3:00 pm to 5:00 pm each day. The thickness of the slide is 1.5 mm and it is evenly coated with a white petroleum jelly film. The slides were changed at the end of each day's test and the slides with adsorbed spores of the original wheat powdery mildew fungus were examined microscopically. For microscopic observation, place a drop of sterile distilled water in the centre of the slide, cover with a coverslip (18 mm × 18 mm) and count the number of powdery mildew protozoa spores in that range using microscopic observation. When the change in the number of powdery mildew conidia in the control and treatment areas levelled off for two consecutive days, wheat was disturbed with the airflow generated by the drone rotors at 3 p.m. each day in the treatment areas (Area A and B), with the N-3 plant protection drone flying a single-stroke flight width of 8 m for a total of five strokes; The HyB-15L plant protection drone has a flight width of 5 m in a single stroke and flies 8 strokes in

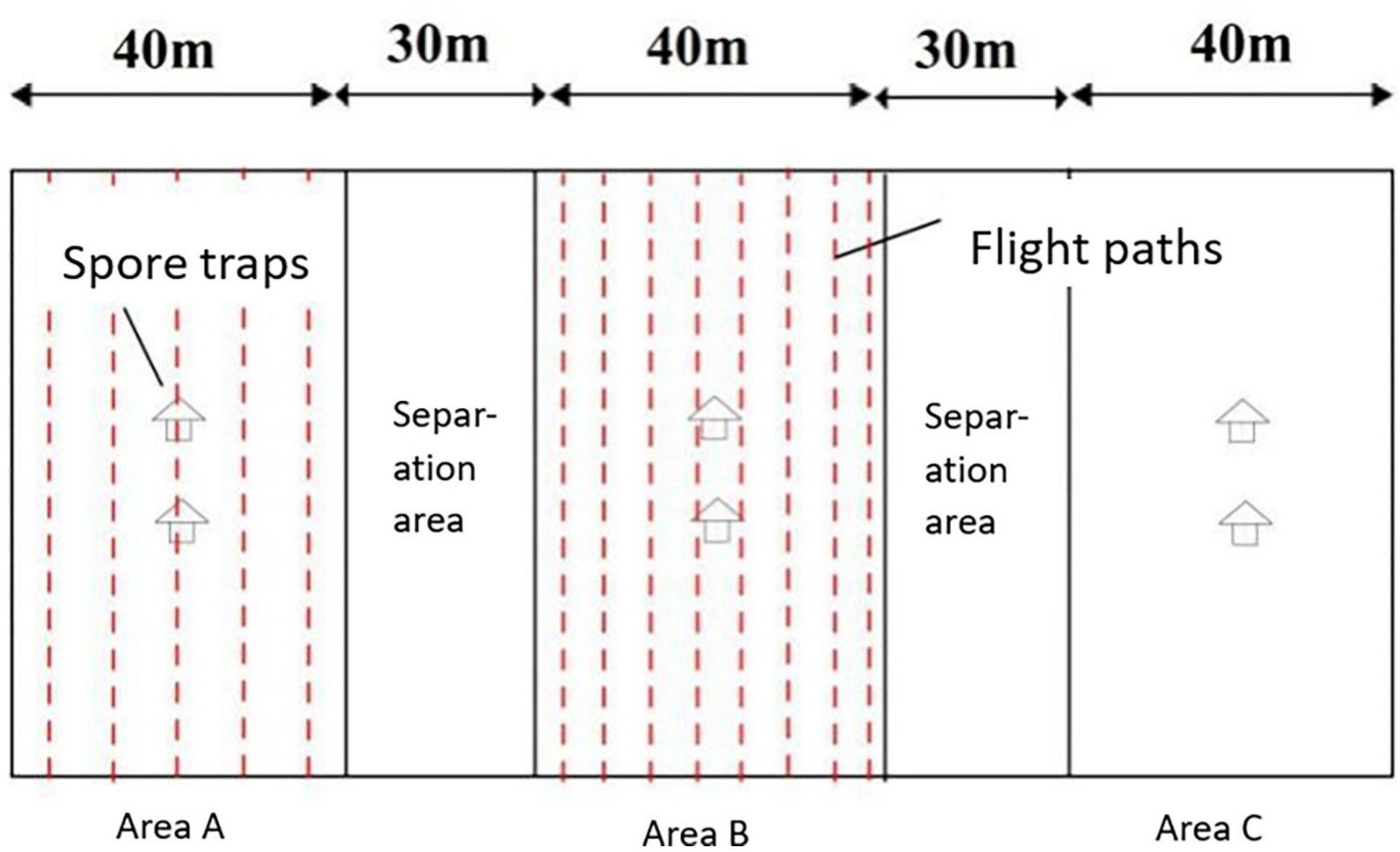


Fig 1. Diagram of test area.


Table 2. The test record of meteorological data.

| Date | Daily rainfall (mm) | Daily maximum temperature (˚C) | Daily minimum temperature (˚C) | Average daily temperature (˚C) | Average daily humidity (%) |
|---|---|---|---|---|---|
| 4.20 | 1.7 | 18.5 | 10.1 | 12.8 | 85.8 |
| 4.21 | 0 | 20.1 | 3.2 | 13.2 | 76.4 |
| 4.22 | 0 | 25.1 | 6.4 | 16.5 | 65.2 |
| 4.23 | 0 | 26.9 | 11.3 | 18.7 | 50.6 |
| 4.24. | 0 | 27.5 | 19.8 | 20.3 | 70.2 |
| 4.25 | 0 | 27.0 | 11.1 | 19.4 | 68.4 |
| 4.26 | 0 | 28.3 | 9.6 | 21.2 | 52.3 |
| 4.27 | 2.8 | 26.2 | 14.5 | 21.2 | 85.1 |
| 4.28 | 11.9 | 24.0 | 14.5 | 18.0 | 90.2 |
| 4.29 | 0 | 21.7 | 14.7 | 17.3 | 74.3 |
| 4.30 | 0 | 25.3 | 9.1 | 18.5 | 63.8 |


total; The comparsion area (Area C) is the undisturbed area. Spores were collected for 3 hours at the end of the test and brought back to the room for observation at the end. Where not stated, conidial values recorded during the experiment are the amounts observed on this area. Equidistant sampling method on each slide [38].

**2.3.3. Environmental monitoring.** Meteorological data for the trial area was provided by the Hongze County Plant Protection Station, including daily maximum and minimum temperatures, humidity and rainfall. The data are shown in Tables 2 and 3:

**2.3.4. Disease investigation.** A modified "0–9" scale method was used to investigate the incidence of disease in the field plots As shown in Table 4. 20 plants were sampled at 5 points per plot and the incidence levels were recorded separately (Disease Index, Abbreviations DI) [39–41].

$$DI = (0 \times n0 + 1 \times n1 + \ldots + 9 \times n9)/(9 \times (n0 + n1 + \ldots + n9)) \times 100$$

where n0, n1. . .. . .n9 represent the number of strains surveyed at different levels of disease, respectively.

**2.3.5. Data analysis.** The data were statistically analysed using SPSS software.

Table 3. The test record of wind speed.

| Date | Daily maximum wind speed ($m \cdot s^{-1}$) | Daily minimum wind speed ($m \cdot s^{-1}$) | Daily average wind speed ($m \cdot s^{-1}$) |
|---|---|---|---|
| 4.20 | 3.2 | 0 | 1.5 |
| 4.21 | 1.8 | 0 | 3.4 |
| 4.22 | 4.1 | 0 | 2.7 |
| 4.23 | 3.6 | 0 | 1.8 |
| 4.24. | 2.8 | 0 | 2.1 |
| 4.25 | 5.6 | 0 | 2.4 |
| 4.26 | 5.1 | 0 | 1.9 |
| 4.27 | 6.8 | 0 | 2.7 |
| 4.28 | 7.4 | 0 | 1.5 |
| 4.29 | 4.7 | 0 | 1.7 |
| 4.30 | 2.5 | 0 | 1.3 |


Table 4. The classification standard of wheat powdery mildew.

| Disease Classification | Description |
|---|---|
| 0 | Full column free of disease. |
| 1 | Few spots of disease on leaves on first segment (Disease spots covering less than 2% of leaf area). |
| 2 | Few spots of disease on leaves on second segment, light leaf disease on the first segment (Disease spots covering 3–10% of leaf area). |
| 3 | Light disease on the third leaf section, moderate disease on the second leaf section (spots covering 11–25% of the leaf area), severe disease on the first leaf section. |
| 4 | Mild disease on fourth segment leaves, moderate to severe disease on third segment and below. |
| 5 | Mild disease on leaves of the fifth segment, moderate to severe disease on leaves of the fourth segment and below. |
| 6 | Mild disease on leaves of the sixth segment, moderate to severe disease on leaves of the fifth segment and below. |
| 7 | Light disease on leaves of section 7, moderate to severe disease on leaves of section 6 and below, sometimes with occasional few spots on flag leaves. |
| 8 | Mild or moderate disease on flag leaf, moderate to severe disease on leaves below flag leaf. |
| 9 | Severe disease on overall leaves, varying degrees of disease on spikes, spike disease expressed as a percentage. |


## 3. Results and analysis

### 3.1. Wheat powdery mildew pathogen spore monitoring results

As Fig 2 shows, the number of spores appearing before 22 April was low. The spores of the wheat powdery mildew pathogen began to appear continuously in the field after 23 April and remained there until 26 April when their number peaked at a cumulative total of 38. While the number of aerial spores decreased on 27 April, which was related to the rainfall that occurred on the 27th and 28th of the day, with 2.8 mm and 11.9 mm of precipitation, respectively. A

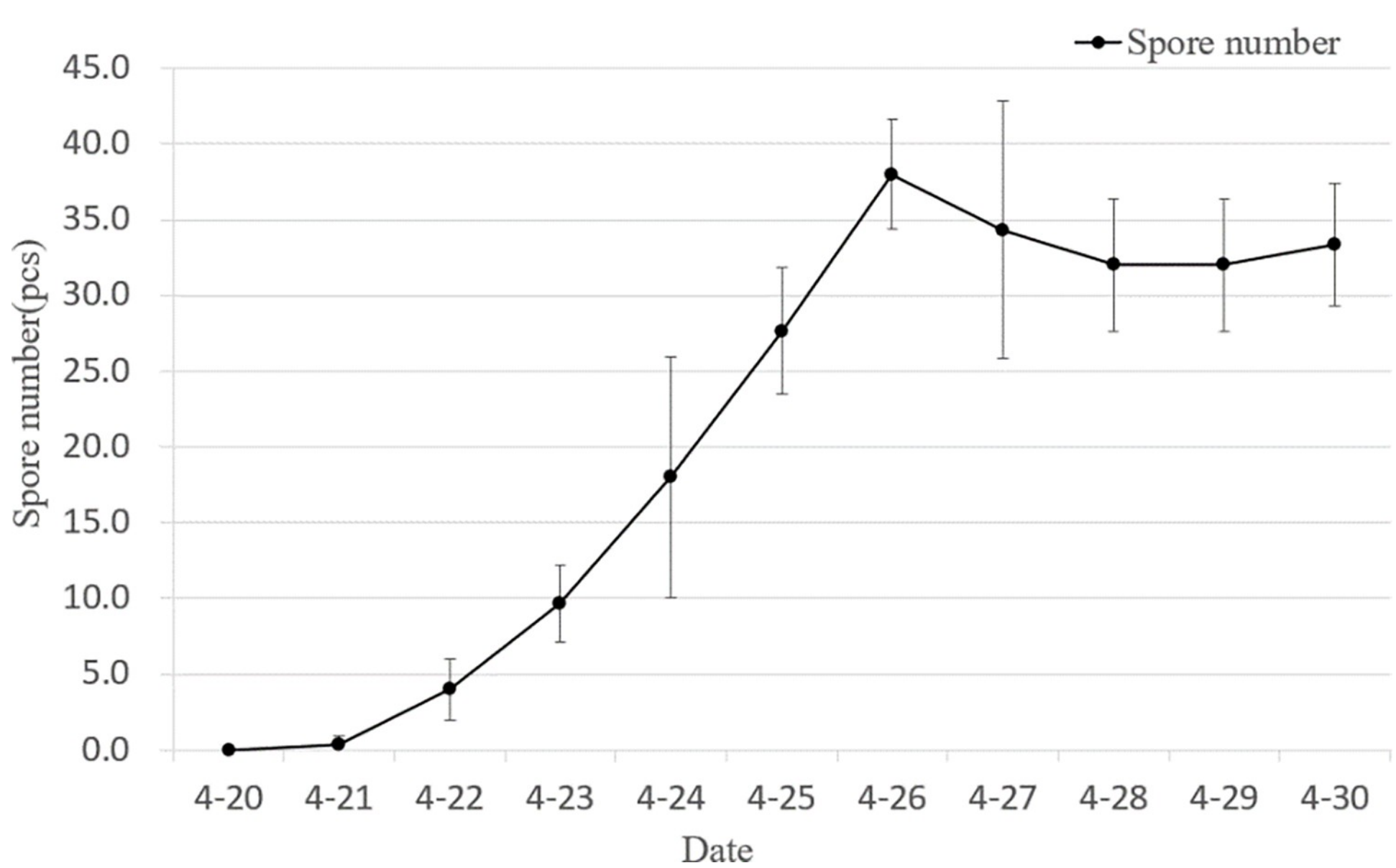


Fig 2. The individuals of conidial concentration of wheat powdery mildew during the test.


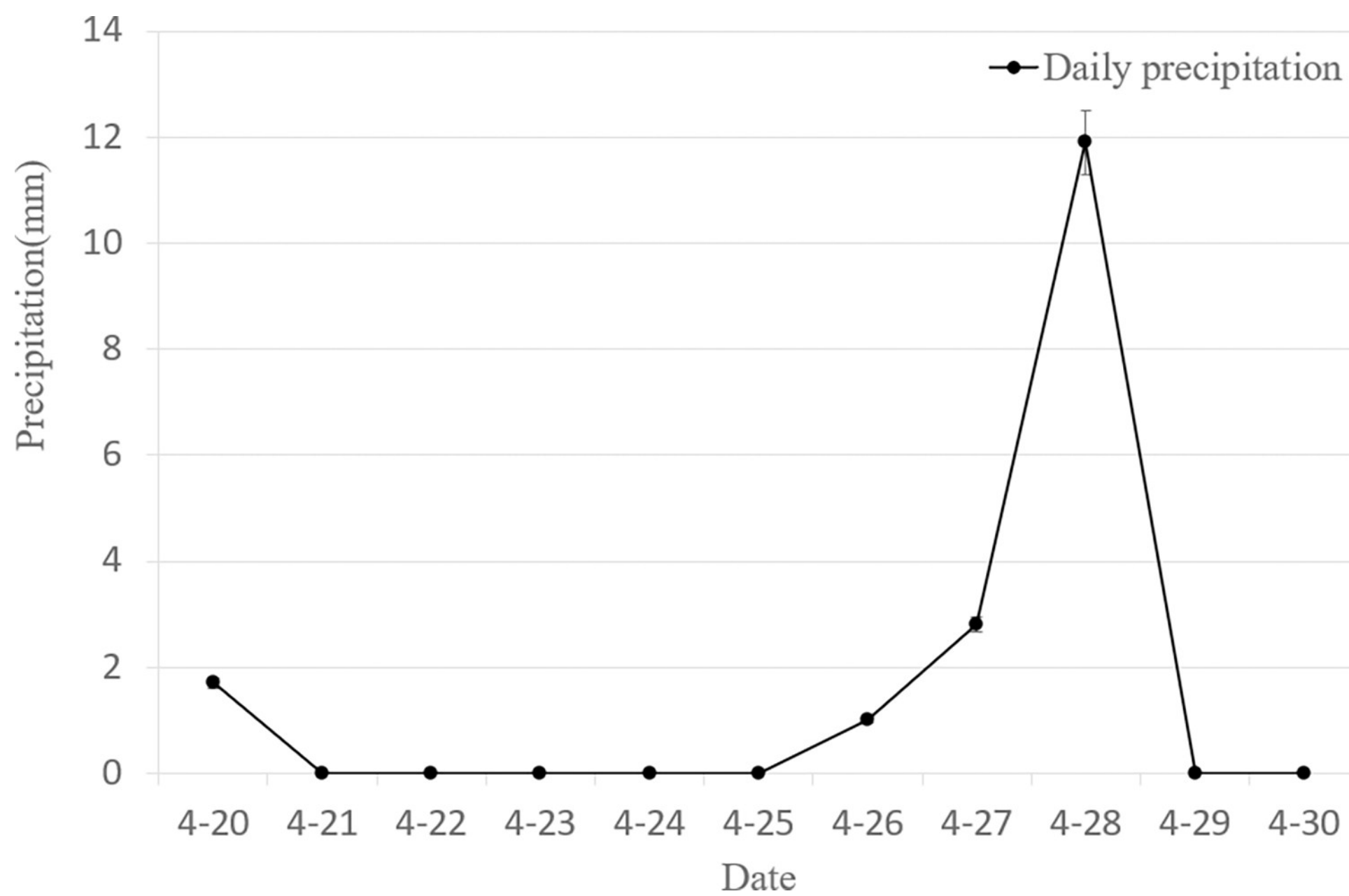


**Fig 3. Dilay precipitation during the test.**


diagram of daily precipitation is shown in Fig 3. After 30 days, there is a trend towards an increase in the number of spores.

As shown in Figs 4 and 5. In terms of temperature and relative humidity, the average daily temperature rose gradually from 20 to 24 April, from 12.8˚C to 20.3˚C, while the relative humidity varied between 50% and 90%. The temperature dropped on the 25th but then trended up again. By the 28th, temperatures have dropped after the rain, with daily temperatures averaging around 16˚C. The average daily relative humidity reached 90% on the day of the rain.

The temperature changes during the observation of sporulation were all below 21˚C, which is in line with the condition that the optimum temperature for the growth of powdery mildew spores is between 15 and 20˚C. Too high a temperature is not conducive to the release of spores. It has been shown that sporulation shows a negative correlation with temperature. As can be seen from Fig 5, there is a highly significant negative correlation between the number of wheat powdery mildew spores and relative humidity [42], with high air humidity being detrimental to the formation and spread of conidia.

There is also an effect of wind speed on the number of powdery mildew spores, as the amount of wind speed affects the distribution and direction of spores in the air. As can be seen from Fig 6, the daily average wind speed during the test period had a maximum value of 3.5 $m \cdot s^{-1}$ and a minimum value of 1.5 $m \cdot s^{-1}$.

## 3.2. Effect of rotor airflow on the release of sporulation of powdery mildew protozoa in wheat

From 21 to 25 April, the amount of spores gradually increased. On the 22nd, the original spores of wheat powdery mildew were captured and recorded again after the simulated spraying flight with the N-3 plant protection drone in area A of the trial, and it can be seen from Fig 7 that from 22 to 25, the amount of spores released increased insignificantly by about 2% compared to the spores in area C without rotor airflow. From 26 onwards, there was a marked

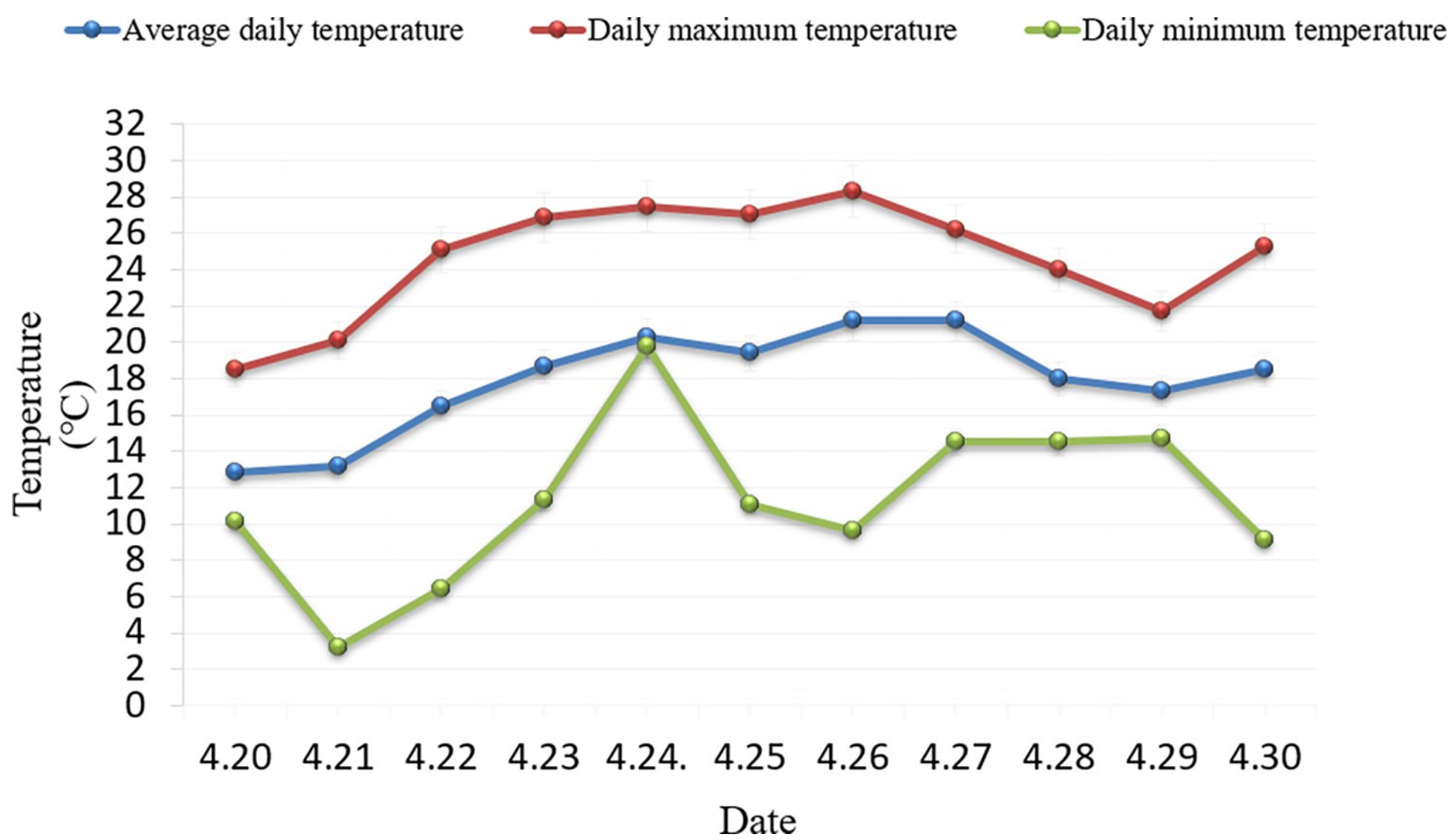


**Fig 4. The temperature of day during test.**


increase in spore release, reaching 66, which is 1.74 times the number of spores in area C without rotor airflow; The same drone simulations were conducted on the 27th and 28th with 63 and 57 spore captures, 1.83 and 1.78 times the number of spores in area C without rotor airflow; The spores of wheat powdery mildew protozoa were captured and recorded after the simulated spraying flight with the HBY-15L plant protection drone in test area B. As can be seen from Fig 8, the number of spores released on the 26th was significantly higher, reaching 56, 1.47 times the number of spores in area C without rotor airflow; The same drone simulations were conducted on the 27th and 28th with 48 and 43 spore captures, 1.41 and 1.34 times the number of spores in area C without rotor airflow. From the above data, it is clear that rotor

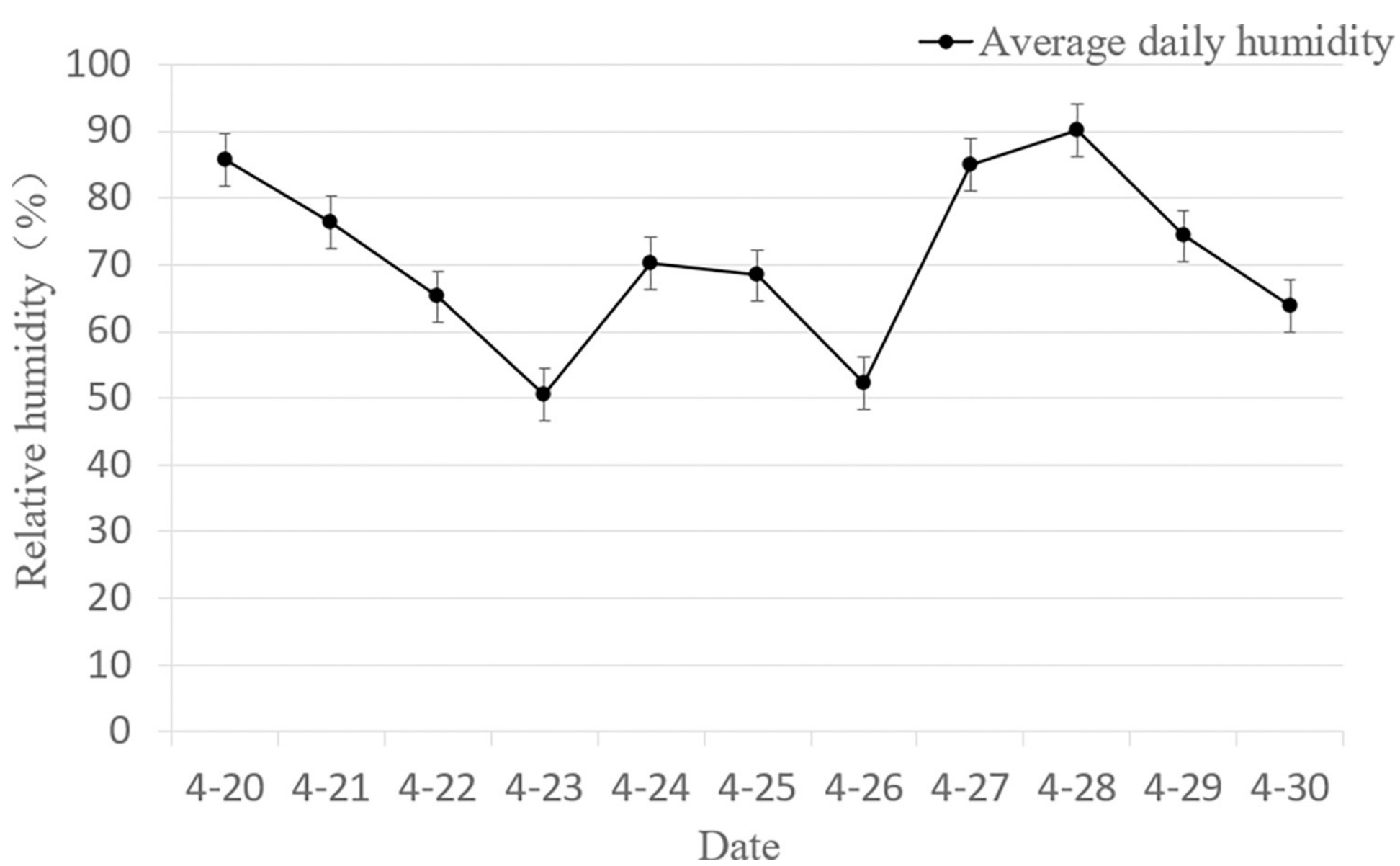


**Fig 5. The average relative humidity of the day during test.**


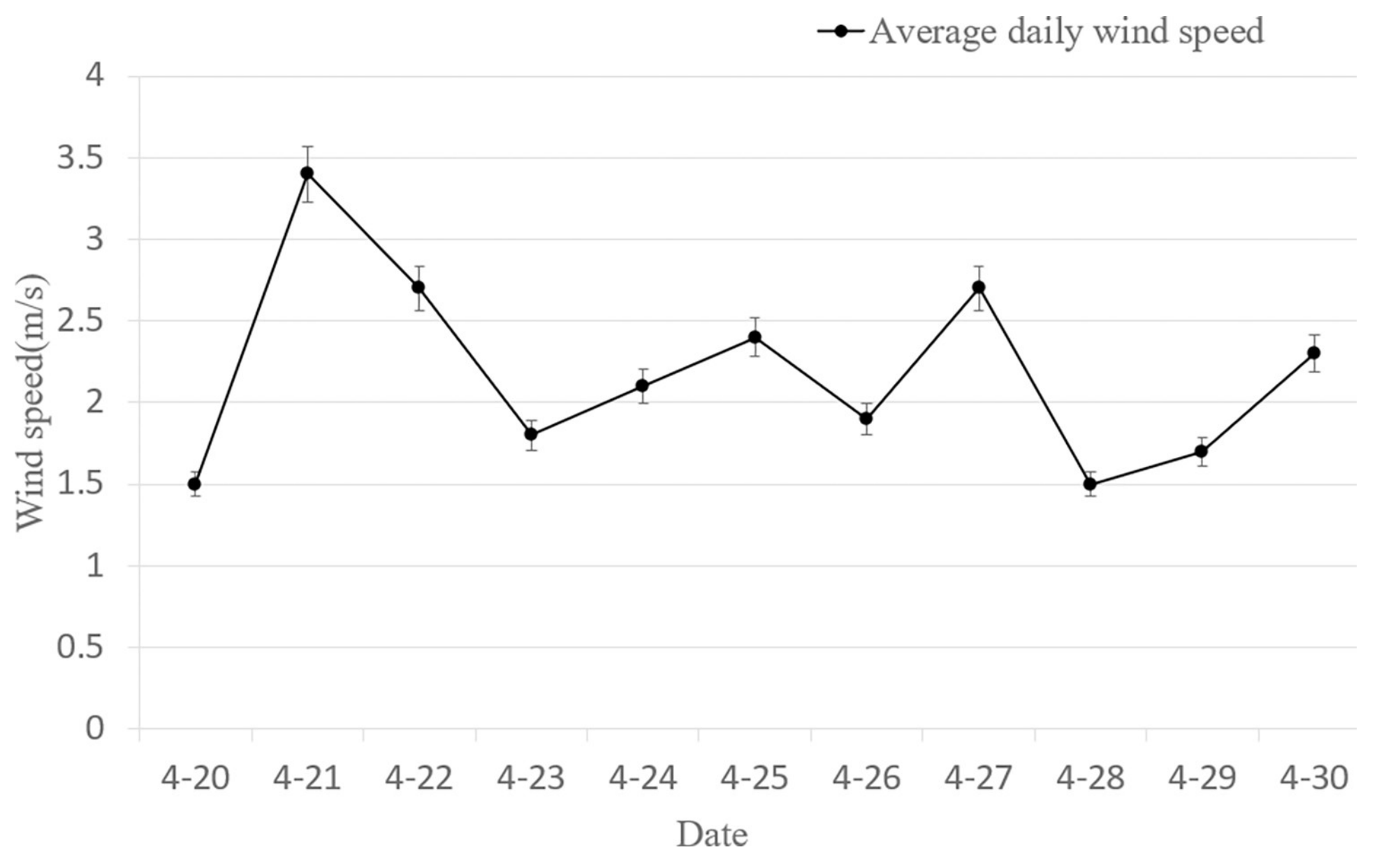


**Fig 6. The average wind speed of the day during test.**


airflow can significantly increase spore release at certain times and that the N-3 plant protection drone is somewhat more effective than the HBY-15L plant protection drone.

### 3.3. Relationship between disease index and cumulative spore count

The disease incidence was surveyed on 30 April and as can be seen from the index of wheat powdery mildew disease in the field and the number of spores accumulated (Fig 9), trial A had the highest number of spores accumulated from 21 to 30 April (294) and the most severe

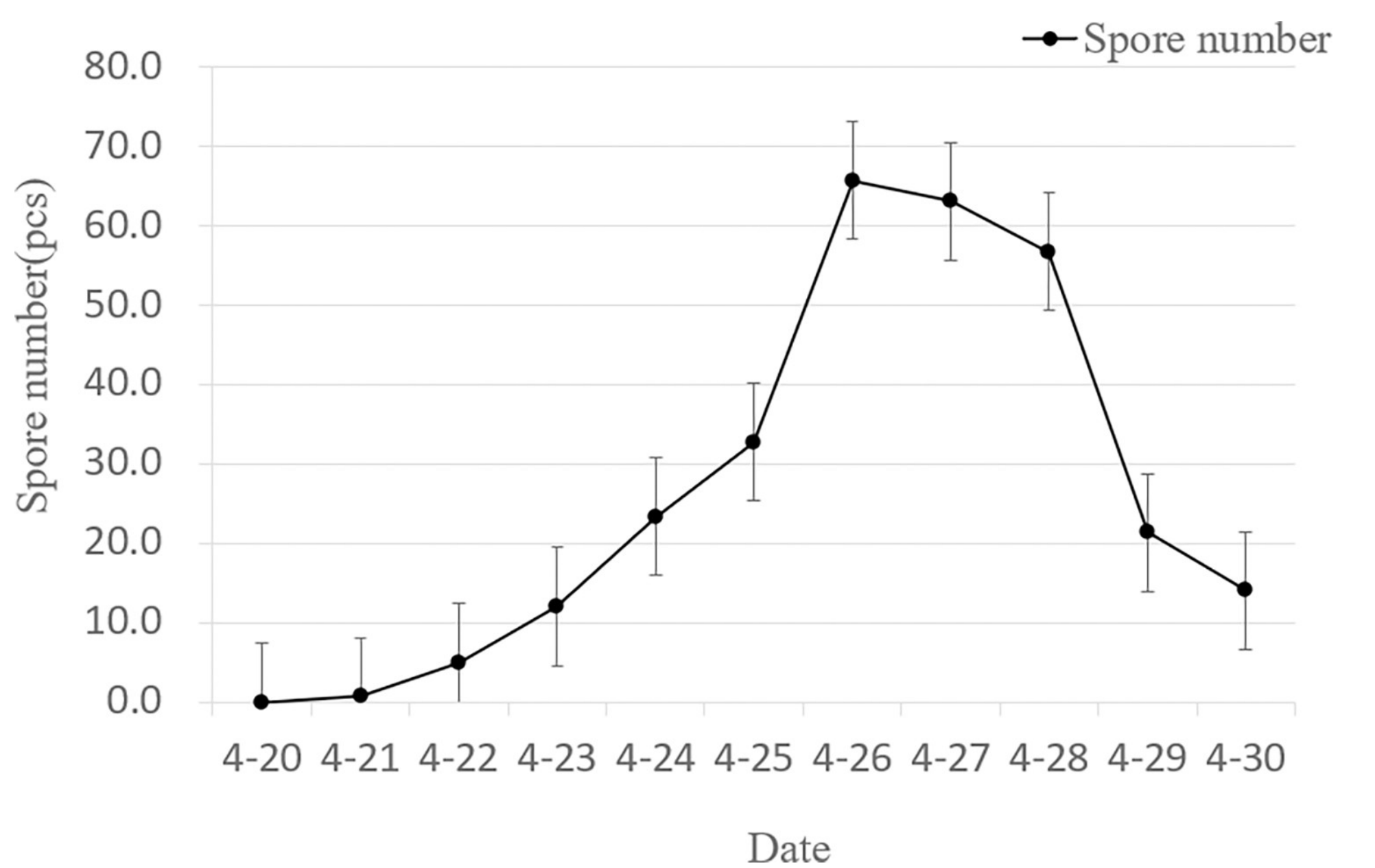


**Fig 7. The effects of N-3 UAV rotor airflow on releasing conidial.**


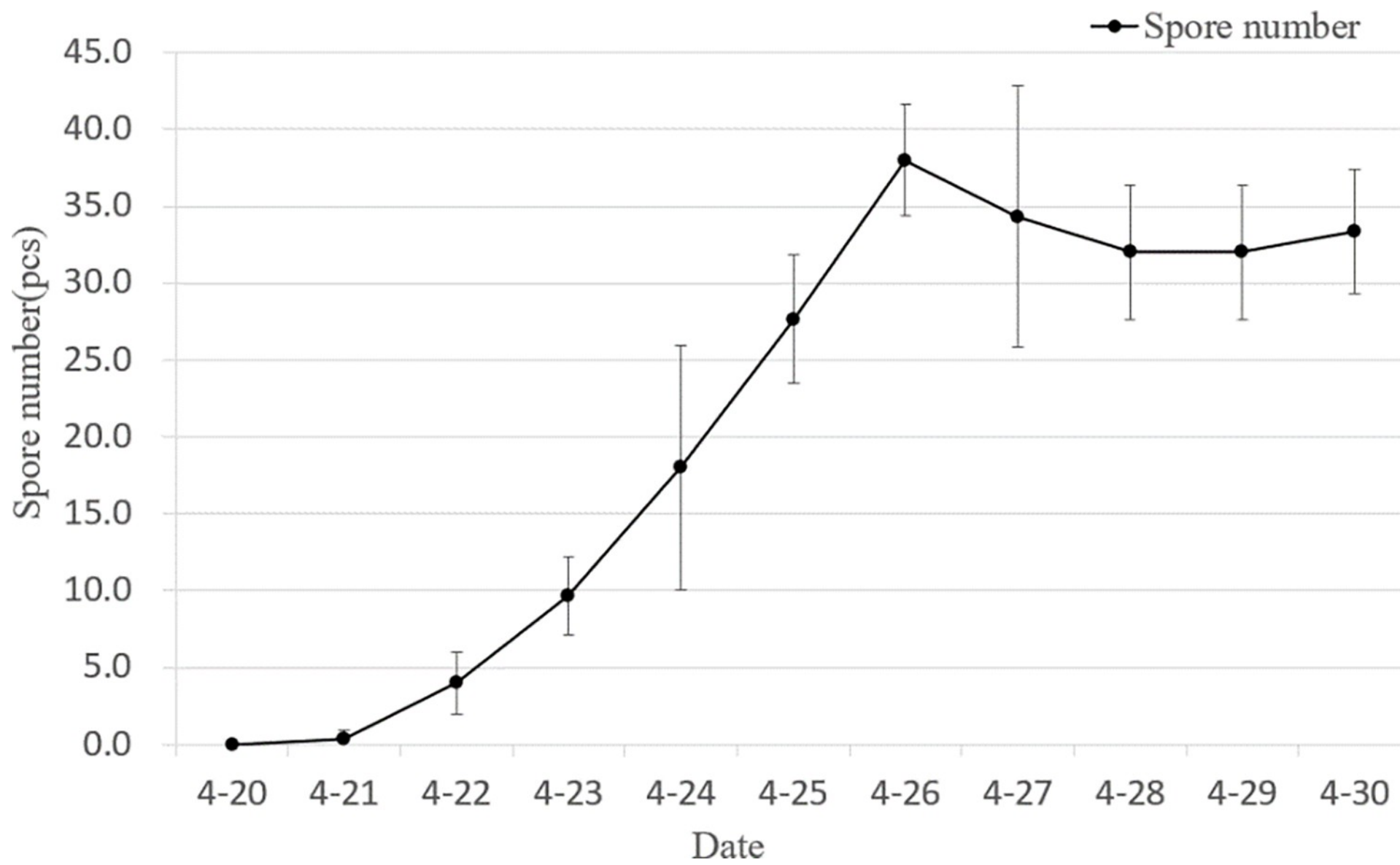


**Fig 8. The effects of HBY-15L UAV rotor airflow on releasing conidial.**


disease incidence, followed by trial B (243) and the lowest was trial C (229). Areas A and B of the trial were affected by the rotor airflow respectively, resulting in a significant increase in the number of spores, making disease incidence 1.45 and 1.28 times higher than in the absence of rotor airflow.

## 4. Conclusion

1. The number of powdery mildew pathogenic spores depends on many factors, related to the actual disease in the field, the amount of fungal sources, airflow disturbances, etc, in addition to meteorological factors (temperature, relative humidity, wind speed, etc).

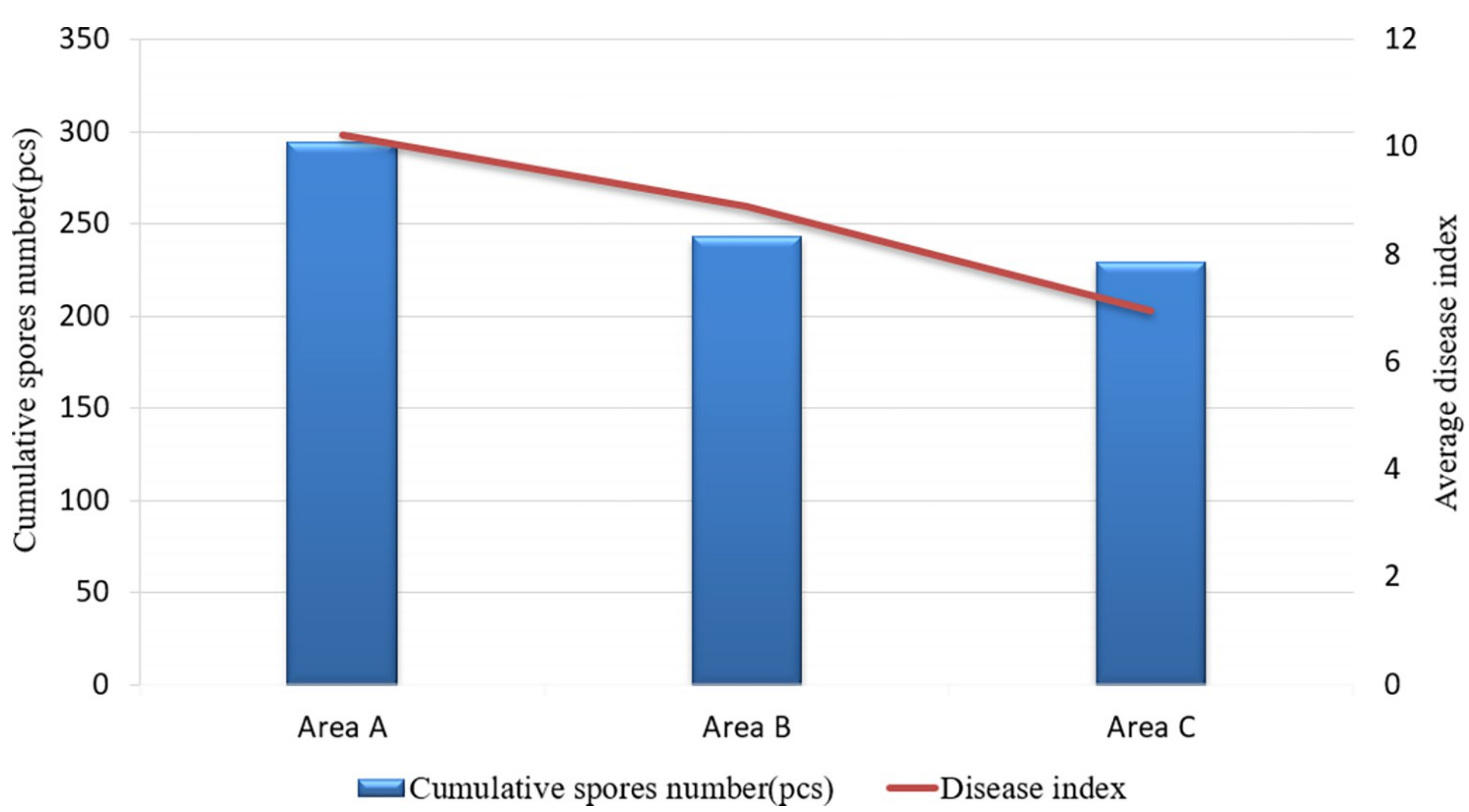


**Fig 9. Coefficients of correlation between cumulative conidials and disease index of wheat powdery mildew.**


2. The release of spore numbers of powdery mildew pathogens is closely related to the effect of airflow disturbance. In the beginning stage when spore release is low, the rotor airflow of the drone has less effect on spore release; as the amount of spores increases, the airflow disturbance of wheat after rotor airflow significantly increases spore release compared to wheat without airflow disturbance, and the larger the rotor diameter, the more extensive the effect.
3. Disease development is a process of accumulation of pathogenic spores, and the more spores that accumulate, the more severe the disease.
4. In wheat powdery mildew drone spraying control, in order to reduce the effect of rotor airflow on spore release, it is recommended to apply preventive spraying control within 2 to 3 days when the pathogen spores first start to appear, and try not to wait until the disease is already evident, as this may increase the incidence of the disease.

## Author Contributions

**Investigation:** Panyang Chen.

**Methodology:** Ruyan He.

**Writing – original draft:** Weicai Qin.

**Writing – review & editing:** Weicai Qin.

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
