## [Decision Letter · Decision Letter 0]

11 May 2023

PONE-D-23-05101Study on the dynamic distribution of spores of powdery mildew pathogen in wheat by rotor airflow of plant protection UAVPLOS ONE

Dear Dr. qin,

Thank you for submitting your manuscript to PLOS ONE. After careful consideration, we feel that it has merit but does not fully meet PLOS ONE’s publication criteria as it currently stands. Therefore, we invite you to submit a revised version of the manuscript that addresses the points raised during the review process.

We look forward to receiving your revised manuscript.

Kind regards,

Abhay K. Pandey

Academic Editor

PLOS ONE

Journal Requirements:

6. Please ensure that you refer to Figure 4 in your text as, if accepted, production will need this reference to link the reader to the figure.

7. We note you have included a table to which you do not refer in the text of your manuscript. Please ensure that you refer to Tables 2, 3 and 4 in your text; if accepted, production will need this reference to link the reader to the Table.

Reviewers' comments:

Reviewer's Responses to Questions

**Comments to the Author**

1. Is the manuscript technically sound, and do the data support the conclusions?

Reviewer #1: Yes

2. Has the statistical analysis been performed appropriately and rigorously? 

Reviewer #1: Yes

3. Have the authors made all data underlying the findings in their manuscript fully available?

Reviewer #1: Yes

4. Is the manuscript presented in an intelligible fashion and written in standard English?

Reviewer #1: Yes

5. Review Comments to the Author

Reviewer #1: 1- Avoid to use uncommon abbreviations in the title.

2- Rewrite Abstract in a more scientific way. Introductory portion in the Abstract is too lengthy. Write Abstract in past tense.

3- Introduction is too lengthy. Reduce its length to 2/3 with maximum of 3 paragraphs.

4- Give spaces between a digit and its units in Figure 1.

5- Discussion is almost missing. Discuss the results in the light of available literature.

6- Format uniformly and correctly.

6. PLOS authors have the option to publish the peer review history of their article (what does this mean?). If published, this will include your full peer review and any attached files.

Reviewer #1: **Yes: **Prof. Dr. Arshad Javaid

---

## [Author Response · Author response to Decision Letter 0]

17 May 2023

Reviewer

1、 Avoid to use uncommon abbreviations in the title.

Answer: The title has been changed to Study on the dynamic distribution of spores of powdery mil-dew pathogen in wheat by rotor airflow of plant protection unmanned aerial vehicles.

2、 Rewrite Abstract in a more scientific way. Introductory portion in the Abstract is too lengthy. Write Abstract in past tense.

Answer: Abstract has been re-edited.

3、 Introduction is too lengthy. Reduce its length to 2/3 with maximum of 3 paragraphs.

Answer: The text of the introduction is amended to read “Wheat powdery mildew caused by the biotrophic fungi Blumeria graminis (DC) Speer f. sp. tritici Emend. E.J. Marshcl, is the most important wheat diseases worldwide, which occurs in regions with maritime and semi-continental climates and causes se-vere yield losses. including in Gansu, Jiangsu Province, China [1,2]. Especially in the heading stage of wheat, powdery mildew occurs seriously in the lower part of the canopy. At this time, the leaves of the wheat canopy cover each other, which makes it difficult for ground equipment to work, or causes great damage to the wheat due to rolling[3-5]. In the later stage of wheat growth, it is difficult to walk and la-bor-intensive when spraying pesticides. Large-volume spraying not only wastes pesti-cides, but also causes serious harm to pesticide application personnel and the ecologi-cal environment [6-8]. Therefore, it is urgent to solve the current situation of low level of mechanization of wheat pest control in China, enhance the ability to prevent and control sudden large-scale pests and diseases, and alleviate the shortage of rural labor [9-13].

Aerial application, commonly called crop dusting, involves spraying crops with fertilizers, pesticides, fungicides, and other crop protection materials from agricultural aircraft [14]. Farmers in Japan and South Korea have relatively small arable land per household, with mountainous terrain and small arable land, which are not suitable for manned fixed-wing aircraft operations [15-18]. Therefore, small unmanned helicopters are the main agricultural aviation in this area [19-23]. In view of the current situation of the development of small unmanned helicopters in China, scholars have carried out an exploration of field spraying experiments [24-27]. For example, Qiu (2013) [28] studied the distribution law of droplet deposition under different flight heights and flight speeds of the CD-10 unmanned helicopter, and established a relationship model between deposition concentration, deposition uniformity, flight speed, flight height and the interaction between the two factors. In agricultural pest control, Xue, et al. (2013) [29] studied the control effect of N-3 unmanned helicopter on rice planthopper and rice leaf roller, pointing out that compared with the traditional field manual ap-plication, the unmanned helicopter operation efficiency can be increased by 60 times, the active ingredient of the spraying liquid can be reduced by 20%-30%, and the labour intensity is greatly reduced, providing a working platform for rapid and effective pre-vention and control of outbreaks of diseases, pests and weeds in paddy fields and promoting the upgrading of large-scale plant protection technology of rice. Qin, et al. (2016) [30] studied the effect of HyB-15L UAV at different working heights and speeds on the deposition and distribution of droplets and the control effect of rice planthop-pers.

The presence of wind is a prerequisite for the spread and dissemination of air-borne diseases, which can directly affect the diffusion distance and rising height of pathogen spores. The release of spores requires not only overcoming its attachment to the spore-forming organ, but also passing through and out of the still air layer of the host's surface. However, the release rate of spores is not linear with the wind speed. Generally speaking, the spores can be released only when the wind speed reaches a certain level. The analysis shows that once the wind speed reaches this speed, almost all the spores that can be released are released around this moment. For example, the direct wind speed of the maize small spot fungus conidia themselves is 5 m·s-1, but be-cause of the interfacial layer effect on the leaf surface, and the closer the leaf surface, the smaller the wind speed, so the wind speed is projected to reach 25 m·s-1 to make the leaf surface wind speed reach 5 m·s-1 (Force equivalent to 2000 times the weight of the spore). However, in practice it was observed that the average wind speed did not reach 25 m·s-1 and that the maize microspot conidia had been released due to gusts of wind (Aylor, 1978) [31,32]. Thus gusts play a greater role than average wind speed in spore release, and the greater the number of gusts, the greater the amount of spores released, showing that gusts play a special role in spore release. However, some pathogens re-quire little force to release, such as powdery mildew, and the release of large numbers of spores can be accelerated when the wind shakes the leaves. In addition, wind has a strong influence on the spread of pathogens, with the amount of wind speed directly related to the distance the pathogens can travel [33]. However, there is a lack of in-depth research on the effect of under-rotor wash airflow on pathogens from plant protection drones. In view of the above situation, this chapter analyzes the influence of the downwash air flow generated by the rotor on the horizontal, vertical and ground distribution of rice canopy rice planthopper and wheat powdery mildew spores by studying the flying height of the plant protection UAV[34-37].”

4、 Give spaces between a digit and its units above.

Answer: Modified.

5、 Discussion is almost missing. Discuss the results in the light of available literature.

Answer: Modified. Some discussion has been added.

6、 References: Format uniformly and correctly.

Answer: Modified.

Journal Requirements

1、 Please ensure that your manuscript meets PLOS ONE's style requirements, including those for file naming.

Answer: The manuscript has been revised according to the template.

2、 In your Methods section, please provide additional information regarding the permits you obtained for the work. Please ensure you have included the full name of the authority that approved the field site access and, if no permits were required, a brief statement explaining why.

Answer: No permits were required for this study as all data were collected from publicly accessible sources or did not involve the manipulation of animals or sensitive habitats. 

3、 Thank you for stating the following financial disclosure:

“The author(s) received no specific funding for this work.” At this time, please address the following queries:

Answer: This research was funded by the National Natural Science Foundation of China (Grant No.31971804); Jiangsu Modern Agricultural Machinery Equipment and Technology Demonstration and Promotion Project (Grant No. NJ2022-17); Suzhou Agricultural Independent Innovation Pro-ject (SNG2022061); Suzhou Agricultural Vocational and Technical College Landmark Achievement Cultivation Project (CG[2022]02); Soft Science Project of Suzhou Association for Science and Technology.

Answer: The funders had no role in study design, data collection and analysis, decision to publish, or preparation of the manuscript.

Answer: No authors received a salary or financial support from the funders for this study.

d) If you did not receive any funding for this study, please state: “The authors received no specific funding for this work.”.

Answer: The authors received no specific funding for this work.

Answer: The content of the cover letter has been amended, please see the attached document.

4、 In your Data Availability statement, you have not specified where the minimal data set underlying the results described in your manuscript can be found. PLOS defines a study's minimal data set as the underlying data used to reach the conclusions drawn in the manuscript and any additional data required to replicate the reported study findings in their entirety. All PLOS journals require that the minimal data set be made fully available. 

Answer: Data Availability Statement: Not applicable.

5、 PLOS requires an ORCID iD for the corresponding author in Editorial Manager on papers submitted after December 6th, 2016. 

Answer: I make sure you have an ORCID iD. javascript:popup_orcidDetail('https://orcid.org','0000-0001-7436-8834').

6、 Please ensure that you refer to Figure 4 in your text as, if accepted, production will need this reference to link the reader to the figure. 

Answer: The manuscript has been modified so that Figure 4 is mentioned in the manuscript

7、 We note you have included a table to which you do not refer in the text of your manuscript. Please ensure that you refer to Tables 2, 3 and 4 in your text; if accepted, production will need this reference to link the reader to the Table. 

Answer: Tables 2, 3 and 4 have been referred to in the manuscript.

---

## [Decision Letter · Decision Letter 1]

22 Jun 2023

Study on the dynamic distribution of spores of powdery mildew pathogen in wheat by rotor airflow of plant protection unmanned aerial vehicles

PONE-D-23-05101R1

Dear Dr. qin,

We’re pleased to inform you that your manuscript has been judged scientifically suitable for publication and will be formally accepted for publication once it meets all outstanding technical requirements.

Kind regards,

Abhay K. Pandey

Academic Editor

PLOS ONE

Additional Editor Comments (optional):

Reviewers' comments:

Reviewer's Responses to Questions

**Comments to the Author**

1. If the authors have adequately addressed your comments raised in a previous round of review and you feel that this manuscript is now acceptable for publication, you may indicate that here to bypass the “Comments to the Author” section, enter your conflict of interest statement in the “Confidential to Editor” section, and submit your "Accept" recommendation.

Reviewer #1: All comments have been addressed

2. Is the manuscript technically sound, and do the data support the conclusions?

Reviewer #1: Yes

3. Has the statistical analysis been performed appropriately and rigorously? 

Reviewer #1: Yes

4. Have the authors made all data underlying the findings in their manuscript fully available?

Reviewer #1: Yes

5. Is the manuscript presented in an intelligible fashion and written in standard English?

Reviewer #1: Yes

6. Review Comments to the Author

Reviewer #1: Authors have addressed all the queries raised by me. Paper has been improved significantly is acceptable in its current form.

7. PLOS authors have the option to publish the peer review history of their article (what does this mean?). If published, this will include your full peer review and any attached files.

Reviewer #1: **Yes: **Prof. Dr. Arshad Javaid, Punjab University Lahore, Pakistan

<quillbot-extension-portal></quillbot-extension-portal>

---

## [Editor Report · Acceptance letter]

28 Jun 2023

PONE-D-23-05101R1 

Study on the dynamic distribution of spores of powdery mildew pathogen in wheat by rotor airflow of plant protection UAV 

Dear Dr. Qin:

I'm pleased to inform you that your manuscript has been deemed suitable for publication in PLOS ONE. Congratulations! Your manuscript is now with our production department. 

Kind regards, 

on behalf of

Dr. Abhay K. Pandey 

Academic Editor

PLOS ONE